# Dehydrocrenatidine Induces Liver Cancer Cell Apoptosis by Suppressing JNK-Mediated Signaling

**DOI:** 10.3390/ph15040402

**Published:** 2022-03-25

**Authors:** Bharath Kumar Velmurugan, Ming-Ju Hsieh, Chia-Chieh Lin, Hsin-Yu Ho, Ming-Chang Hsieh

**Affiliations:** 1Faculty of Applied Sciences, Ton Duc Thang University, Ho Chi Minh City 756000, Vietnam; bharath.kumar.velmurugan@tdtu.edu.vn; 2Oral Cancer Research Center, Changhua Christian Hospital, Changhua 500, Taiwan; 170780@cch.org.tw (M.-J.H.); 181327@cch.org.tw (C.-C.L.); 183581@cch.org.tw (H.-Y.H.); 3Department of Post-Baccalaureate Medicine, College of Medicine, National Chung Hsing University, Taichung 402, Taiwan; 4Graduate Institute of Biomedical Sciences, China Medical University, Taichung 404, Taiwan; 5School of Medical Laboratory and Biotechnology, Chung Shan Medical University, Taichung 40201, Taiwan; 6Department of Clinical Laboratory, Chung Shan Medical University Hospital, Taichung 40201, Taiwan

**Keywords:** dehydrocrenatidine, liver, apoptosis, JNK

## Abstract

Liver cancer is a leading cause of death worldwide. Despite advancement in therapeutic interventions, liver cancer is associated with poor prognosis because of highly lethal characteristics and high recurrence rate. In the present study, the anticancer potential of a plant-based alkaloid namely dehydrocrenatidine has been evaluated in human liver cancer cells. The study findings revealed that dehydrocrenatidine reduced cancer cell viability by arresting cell cycle at G2/M phase and activating mitochondria-mediated and death receptor-mediated apoptotic pathways. Specifically, dehydrocrenatidine significantly increased the expression of extrinsic pathway components (FAS, DR5, FADD, and TRADD) as well as intrinsic pathway components (Bax and Bim L/S) in liver cancer cells. In addition, dehydrocrenatidine significantly increased the cleavage and activation of PARP and caspases 3, 8, and 9. The analysis of upstream signaling pathways revealed that dehydrocrenatidine induced caspase-mediated apoptosis by suppressing the phosphorylation of JNK1/2. Taken together, the study identifies dehydrocrenatidine as a potent anticancer agent that can be use clinically to inhibit the proliferation of human liver cancer cells.

## 1. Introduction

Hepatocellular carcinoma, the most prevalent form of liver cancer, is a leading cause of cancer-related deaths worldwide. The major risk factors include non-alcoholic fatty liver disease, alcoholic liver disease, steatohepatitis, and hepatis B and/or C virus infection [1]. In Taiwan, about 62% of men and 37% of women have been identified to have liver cancer due to viral hepatitis between 2002 and 2010 [2]. The incidence of liver cancer in men over the age of 40 years is widespread globally, causing nearly 800,000 deaths every year [3]. More than 90% of liver cancer cases are caused by chronic hepatitis B or C infection or liver diseases caused by long-term alcoholism and liver cirrhosis [4]. Despite the reduction in chronic liver disease and liver cancer incidence after 2003 due to the progression of national viral hepatitis treatment plan, a high recurrence rate and high lethal characteristics are still important issues in recent years in Taiwan [5,6]. Surgeries including liver resection and liver transplantation remain the gold standard for treating hepatocellular carcinoma patients. Curative therapies are applied to less than 20% of patients because of the diagnosis of cancer at advanced stages [7]. Although chemotherapy and radiotherapy are used for most of the patients, these palliative treatments are mostly resistant to hepatocellular carcinoma [7]. Thus, it is of prime importance to identify new anticancer agents that can potently inhibit the growth of liver cancer cells.

Natural plant compounds as effective chemotherapeutic agents have gained immense importance in recent years. These compounds are better tolerated than synthetic drugs and have much less side-effects [8,9,10]. In the past few decades, several bioactive plant compounds, including alkaloids, have been identified to have anticancer and anti-proliferative effects [11,12]. Dehydrocrenatidine isolated from a Chinese medicinal plant *Picrasma quassioides* is a beta-carboline alkaloid that is known to exert analgesic effect by suppressing neuronal excitability [13]. Besides analgesic effects, dehydrocrenatidine has been identified as a specific inhibitor of the JAK–STAT pathway and inducer of apoptosis of solid tumors with constitutively activated JAK2 [14]. A recent study has demonstrated that dehydrocrenatidine inhibits proliferation and induces apoptosis of oral squamous carcinoma cells by activating ERK and JNK signaling pathways [15]. The present study was designed to investigate the anticancer and anti-proliferative effects of dehydrocrenatidine in human liver cancer cells.

## 2. Results

### 2.1. Effect of Dehydrocrenatidine on Cell Viability in Liver Cancer Cells

The cytotoxic effect of dehydrocrenatidine was assessed by treating liver cancer cells (Huh-7 and Sk-hep-1 cells) with different doses of dehydrocrenatidine (5, 10 and 20 μM) at different time points (24, 48, and 72 h). The analysis of cell viability revealed that higher doses of dehydrocrenatidine significantly reduced the proliferation of liver cancer cells in a time-dependent manner (Figure 1B,C). Similarly, a dose-dependent reduction in colony formation ability was observed in dehydrocrenatidine-treated cells (Figure 1D).

### 2.2. Effect of Dehydrocrenatidine on Cell Cycle Progression in Liver Cancer Cells

To understand the anti-proliferative mode of action of dehydrocrenatidine, flow cytometric analysis was conducted to investigate the effects of different concentrations (5, 10 and 20 μM) of dehydrocrenatidine on the cell cycle progression of liver cancer cells. The findings revealed that the dehydrocrenatidine treatment of liver cancer cells caused significant cell cycle arrest at the G2/M phase, and decreased the cell numbers at G0/G1 phase. The distribution of cells at G2/M phase could reach 70.3% and 71% at high concentration of dehydrocrenatidine in both Huh-7 and Sk-hep-1 cells, respectively (Figure 2).

### 2.3. Effect of Dehydrocrenatidine on Apoptosis of Liver Cancer Cells

Given the profound effect of dehydrocrenatidine on cell cycle progression, further experiments were conducted to determine the impact of dehydrocrenatidine treatment on apoptotic cell death. The liver cancer cells were treated with different concentrations of dehydrocrenatidine and subjected to DAPI staining and fluorescence microscopic analysis to visualize chromatin condensation. In addition, dehydrocrenatidine-treated cells were subjected to annexin-V/PI double staining and flow cytometric analysis to determine the percentage of apoptotic cells. The findings revealed that the dehydrocrenatidine treatment caused a significant induction in chromatin condensation in liver cancer cells in a dose-dependent manner (Figure 3A,B). Chromatin condensation together with DNA fragmentation is considered as the major hallmark of apoptotic cells [16]. Thus, these findings clearly indicate the pro-apoptotic effect of dehydrocrenatidine on liver cancer cells. Besides chromatin condensation, the dehydrocrenatidine treatment significantly increased the percentage of apoptotic cells in a dose-dependent manner (Figure 3C,D), further justifying the impact of dehydrocrenatidine on apoptotic cell death pathways in liver cancer cells.

### 2.4. Pro-Apoptotic Mode of Action of Dehydrocrenatidine in Liver Cancer Cells

Given the pronounced pro-apoptotic impact of dehydrocrenatidine on liver cancer cells, further experiments were conducted to investigate the mode of action of dehydrocrenatidine in inducing apoptosis. The liver cancer cells were treated with different concentrations of dehydrocrenatidine and subjected to flow cytometric analysis to determine the mitochondrial membrane potential. In addition, proteins extracted from treated cells were subjected to Western blot analysis to determine the expressions of apoptotic pathway components. The findings of flow cytometry revealed that the dehydrocrenatidine treatment caused a significant induction in mitochondrial membrane depolarization in a dose-dependent manner (Figure 4A,B). The Western blot findings revealed that the treatment of dehydrocrenatidine significantly increased the expression of death receptors (Fas and DR5) and other death domain-containing adaptor proteins (FADD and TRADD) in Huh-7 cells in a dose-dependent manner (Figure 4C,D). Moreover, the expression of death receptor associated proteins also found elevating in Sk-hep-1 cells after high dose of dehydrocrenatidine (20 µM) treatment. The Western blot analysis of intrinsic apoptotic pathway components revealed that the dehydrocrenatidine treatment significantly increased the expressions of cleaved caspases 3, 8, and 9 and cleaved PARP (Figure 5A,B). In addition, the dehydrocrenatidine treatment significantly increased the expressions of pro-apoptotic proteins Bax and Bim L/S and significantly reduced the expressions of anti-apoptotic proteins Bcl-xL and Bcl-2 (Figure 5C,D). Taken together, these findings indicate that dehydrocrenatidine exerts pro-apoptotic effects by inducing both intrinsic and extrinsic pathways in liver cancer cells.

### 2.5. Pro-Apoptotic Mechanism of Dehydrocrenatidine in Liver Cancer Cells

The MAPK signaling pathways play a dual role in regulating apoptosis. Depending upon the types of cells and stimuli, the components of the MAPK pathways (ERK1/2, p38, and JNK1/2) either induce or inhibit the apoptotic signaling [17]. In addition, the PI3K-AKT signaling pathway plays a vital role in tumorigenesis by inhibiting apoptosis and promoting cell survival [18]. To identify the upstream signaling pathways perturbed by dehydrocrenatidine, the phosphorylation status of ERK1/2, p38, JNK1/2, and AKT was determined in dehydrocrenatidine-treated liver cancer cells. The findings revealed that dehydrocrenatidine significantly and dose-dependently reduced the phosphorylation of JNK1/2 in liver cancer cells. However, no significant impact of dehydrocrenatidine treatment was observed on the phosphorylation of other tested proteins (Figure 6A,B). To further confirm the involvement of JNK1/2 signaling, the cells were pretreated with JNK inhibitor SP600125, followed by dehydrocrenatidine treatment. In these co-treated cells, the expressions of cleaved PARP and cleaved caspases 3 and 8 were assessed. The findings revealed that the co-treated cells had significantly higher expressions of all tested proteins compared to the cells treated with dehydrocrenatidine alone (Figure 6C,D). The co-treatment with dehydrocrenatidine and another specific JNK inhibitor, JNK-in-8, also found the similar results (Figure 6E,F). These findings confirm that dehydrocrenatidine induces cell death in liver cancer by suppressing the phosphorylation of JNK1/2.

## 3. Discussion

The present study was designed to evaluate the anticancer effect of dehydrocrenatidine, a beta-carboline alkaloid isolated from *Picrasma quassioides,* in human liver cancer. Previous studies investigating therapeutic benefits of dehydrocrenatidine have demonstrated that this natural plant compound acts as a specific inhibitor of the JAK2 signaling pathway and that it inhibits the growth of breast and prostate cancer cells by targeting the JAK–STAT pathway [14]. Apart from exerting analgesic effect by inhibiting neuronal excitability [13], dehydrocrenatidine has been found to exert anti-proliferative effects on different types of cancers, including hepatocellular, nasopharyngeal, head and neck, and oral squamous cell carcinomas [15,19,20,21].

As observed in this study, dehydrocrenatidine dose-dependently reduces the viability and colony formation ability of human liver cancer cells (Figure 1). In addition, dehydrocrenatidine has been found to cause cell cycle arrest at the G2/M phase (Figure 2), as well as induce apoptotic cell death in human liver cancer cells (Figure 3). Regarding mode of action, dehydrocrenatidine has been found to activate both mitochondria-mediated intrinsic and death receptor-mediated extrinsic apoptotic pathways to affect the viability of liver cancer cells (Figure 4 and Figure 5).

As observed in Figure 4, dehydrocrenatidine disrupts mitochondrial functions by increasing mitochondrial membrane depolarization. These findings are in line with a recently published study on hepatocellular carcinoma where dehydrocrenatidine has been shown to induce apoptosis by targeting mitochondrial complexes I, III, and IV and subsequently dysregulating mitochondrial functionality [19]. In another study on hepatocellular carcinoma, dehydrocrenatidine has been shown to prevent migration and invasion of cancer cells by suppressing the epithelial-to-mesenchymal transition [22]. In addition, dehydrocrenatidine has been found to exert cytotoxic effects on nasopharyngeal carcinoma and oral cancer cells by inducing cell cycle arrest, altering mitochondrial membrane potential, and triggering both intrinsic and extrinsic apoptotic pathways [15,20]. These findings are consistent with the observations made in the present study. Interestingly, we found that dehydrocrenatidine induced Huh-7 cell apoptosis by both extrinsic pathway (Fas, DR5, FADD and TRADD) and intrinsic pathway (Bax, Bim L/S and Bcl-2). However, the effect of dehydrocrenatidine on Sk-hep-1 was mainly via intrinsic pathway (Bax, Bim L/S Bcl-xL and Bcl-2), and increasing on extrinsic pathway associated proteins (Fas, DR5, FADD and TRADD) at high dose of dehydrocrenatidine. The difference between dehydrocrenatidine-affected apoptosis pathway might because of the inter-individual differences of cell lines.Regarding dehydrocrenatidine-mediated alteration in upstream signaling pathways, the study findings reveal that the compound mediates its pro-apoptotic effect by suppressing the phosphorylation of JNK1/2 (Figure 6). Compared to dehydrocrenatidine treatment alone, a significantly higher induction of apoptosis has been observed in liver cancer cells treated with both dehydrocrenatidine and JNK1/2 inhibitor. These observations are in line with a recently published study wherein dehydrocrenatidine has been shown to induce cell apoptosis by inhibiting JNK1/2 phosphorylation [20]. Similarly, another recently published study has shown that dehydrocrenatidine prevents head and neck cancer cell motility, migration, and invasion by inhibiting ERK1/2 and JNK1/2 phosphorylation and reducing MMP-2 expression [21].

In contrast to above-mentioned observations, one recently published study on oral squamous cell carcinoma has shown that dehydrocrenatidine mediates pro-apoptotic effects by inducing ERK1/2 and JNK1/2 phosphorylation. This differential response could be due to significantly higher concentrations of dehydrocrenatidine (0, 25, 50, and 100 μM) used in that study. The use of very high concentrations of drugs in in vitro experiments is known to have off-target effects, justifying the discrepancies between study findings [23].

## 4. Materials and Methods

### 4.1. Reagents and Chemicals

Dehydrocrenatidine (≥98% purity) was purchased from ChemFaces (CheCheng Rd, Wuhan, PRC). The stock solution (100 mM) was prepared by dissolving dehydrocrenatidine in dimethyl sulfoxide (DMSO) and kept at −20 °C for further use. The JNK-specific inhibitor (SP600125) was purchased from Santa Cruz Biotechnology (Santa Cruz, CA, USA), and a stock solution of 20 mM was prepared using DMSO. Another JNK-specific inhibitor (JNK-in-8) was purchased from Selleck Chemicals (Houston, TX, USA), and a stock solution of 5 µM was prepared using DMSO. The primary and secondary antibodies were purchased from Cell Signaling Technology (Danvers, MA, USA). The phenylindole (DAPI) dye and 3-(4,5-dimethylthiazol-2-yl)-2,5-diphenyltetrazolium bromide (MTT) were purchased from Sigma-Aldrich (St Louis, MO, USA), and stock solutions were prepared at a concentration of 10 mg/mL and 5mg/mL, respectively.

### 4.2. Cell Culture

Two human hepatoma cell lines Huh-7 and Sk-hep-1 were obtained from the Food Industry Research and Development Institute (Hsinchu, Taiwan). The cells were cultured in Dulbecco’s modified Eagle’s medium (DMEM) (Gibco BRL, Grand Island, NY, USA) supplemented with 10% fetal bovine serum (FBS), 1 mM glutamine, and 1% penicillin/streptomycin. The cells were maintained at 37 °C in a humidified atmosphere containing 5% CO_2_.

### 4.3. MTT Assay

Huh-7 and Sk-hep-1 cells were seeded onto 96-well plates with 1 × 10^4^ cells per well and subsequently treated with different concentrations of dehydrocrenatidine (5, 10 and 20 μM) for 24, 48 and 72 h. The untreated cells were used as controls. After the treatment, MTT stock solution mixed with culture medium was added to each well, followed by recultivation of cells in the incubator for 4 h. The blue crystallized products generated during the reaction was dissolved with methanol. The color intensity was measured at 595 nm. wavelength.

### 4.4. Colony-Formation Assay

Huh-7 and Sk-hep-1 cells were seeded onto 6-well plates with 1 × 10^3^ cells per well, and subsequently treated with 0, 5, 10 and 20 μM of dehydrocrenatidine for 10 days. After the treatment, the colonies were fixed with methanol and visualized using Giemsa (MilliporeSigma, Burlington, MA, USA) staining. Finally, the colonies were photographed and counted for the statistical analysis.

### 4.5. Cell Cycle Analysis

Huh-7 and Sk-hep-1 cells were treated with 0, 5, 10 and 20 μM of dehydrocrenatidine for 24 h, and subsequently fixed with 75% ethanol. The fixed cells were kept at frozen condition overnight (16–18 h) prior to staining with propidium iodide (PI). The cell cycle distribution was analyzed for 5000 collected cells using Muse^®^ Cell Analyzer Assays (Millipore, Burlington, MA, USA).

### 4.6. DAPI Staining

Huh-7 and Sk-hep-1 cells were treated with 0, 5, 10 and 20 μM of dehydrocrenatidine, followed by fixing with 4% paraformaldehyde and staining with DAPI reagent (1:10,000, in triton-X100 mixed with 1X PBS) in the dark for 30 min. Finally, the cells were photographed using fluorescence microscopy (Lecia, Bensheim, Germany). The chromosome condensation and fragmentation were observed and quantified.

### 4.7. Annexin V/PI Double Staining Assay

Huh-7 and Sk-hep-1 cells were seeded onto 6-well plates with 4 × 10^5^ cells per well, and subsequently treated with 0, 5, 10 and 20 μM of dehydrocrenatidine for 24 h. Then, Muse Annexin V & Dead Cell Assay Kit (Millipore) was used to quantify 5000 collected cells in different stages of cell death. The data was collected using Muse^®^ Cell Analyzer Assays (Millipore).

### 4.8. Mitochondrial Membrane Potential Assay

As previously described [24], a Muse Mitochondrial membrane potential assay Kit (Merck Millipore) was used for the experiment. Huh-7 and Sk-hep-1 cells were treated with different concentrations of dehydrocrenatidine (0, 5, 10 and 20 μM) for 24 h, followed by staining with Muse MitoPotential dye for 20 min at 37 °C. The data was obtained using the Muse^®^ Cell Analyzer Assays (Merck Millipore) of 5000 collected cells.

### 4.9. Protein Extraction and Western Blotting Analysis

Huh-7 and Sk-hep-1 cells were seeded onto 6 cm dishes with 5 × 10^5^ cells per dish, and subsequently treated with 0, 5, 10 and 20 μM of dehydrocrenatidine at 37 °C for 24 h. Next, proteins were extracted from the cells using RIPA buffer containing a mixture of protease inhibitors and phosphatase inhibitors (Millipore Sigma, St. Louis, MO, USA). The proteins were subjected to SDS-PAGE electrophoresis and subsequently transferred onto 0.22 mm polyvinylidene difluoride membranes (PVDF, Millipore Sigma). The membranes were blocked with 5% non-fat milk for 1 h and incubated with indicated primary antibodies overnight at 4 °C. Afterwards, the corresponding horseradish peroxidase-conjugated secondary antibodies were added to the membranes, followed by detection of the signal using chemiluminescent (ECL). Finally, the membranes were photographed with a chemiluminescence fluorescence Image Quant LAS 4000 (GE Healthcare, Berlin, Germany) biomolecule imaging system.

## 5. Conclusions

The study identifies dehydrocrenatidine as a potent inducer of apoptosis in human liver cancer cells. Specifically, the compound reduces liver cancer cell viability by arresting cell cycle and inducing caspase-mediated apoptosis. Regarding mode of action, dehydrocrenatidine induces intrinsic apoptotic pathways by modulating mitochondrial membrane potential. In addition, it induces extrinsic apoptotic pathways by activating death receptors and associated adaptor proteins. Overall, dehydrocrenatidine exerts its pro-apoptotic effects by reducing the phosphorylation of JNK1/2.

## Figures and Tables

**Figure 1 pharmaceuticals-15-00402-f001:**
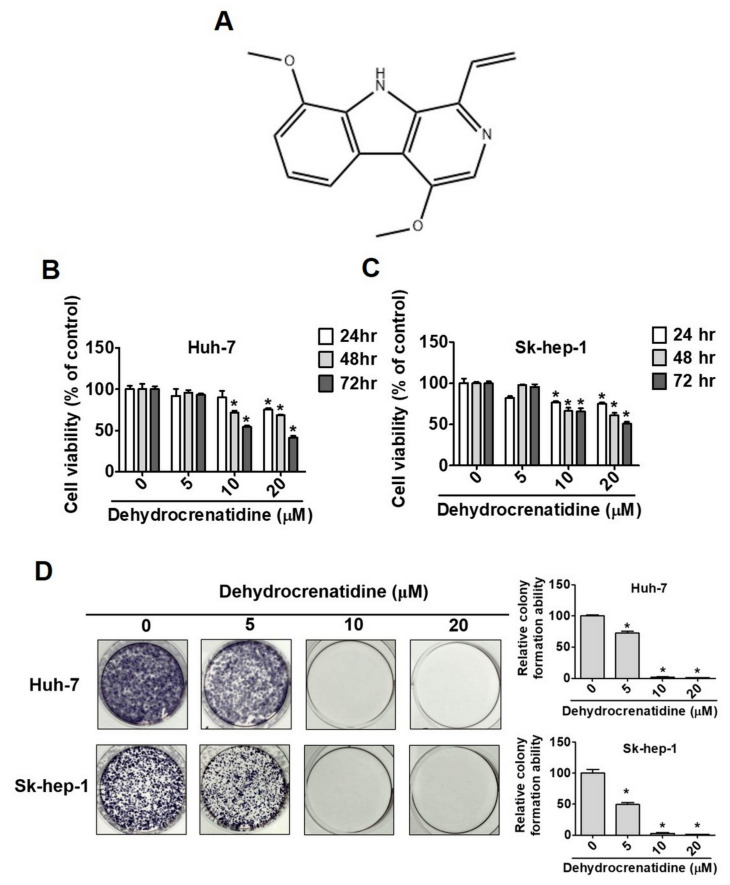
Dehydrocrenatidine induces cytotoxicity and inhibits in vitro cell survival in hepatoma cell lines. (**A**) The chemical structure of dehydrocrenatidine. The cell viability of (**B**) Huh-7 and (**C**) Sk-hep-1 were treated with indicated concentrations of dehydrocrenatidine (0, 5, 10 and 20 μM) for 24, 48, and 72 h. (**D**) Cells that treated with indicated concentrations of dehydrocrenatidine (0, 5, 10 and 20 μM) for 10 days were stained by Giemsa staining buffer. The formazan crystals of Huh-7 and Sk-hep-1 cells were quantified. Student’s *t* test was applied to determine statistical significance in experiment. * represented *p* < 0.05, compared with the control group.

**Figure 2 pharmaceuticals-15-00402-f002:**
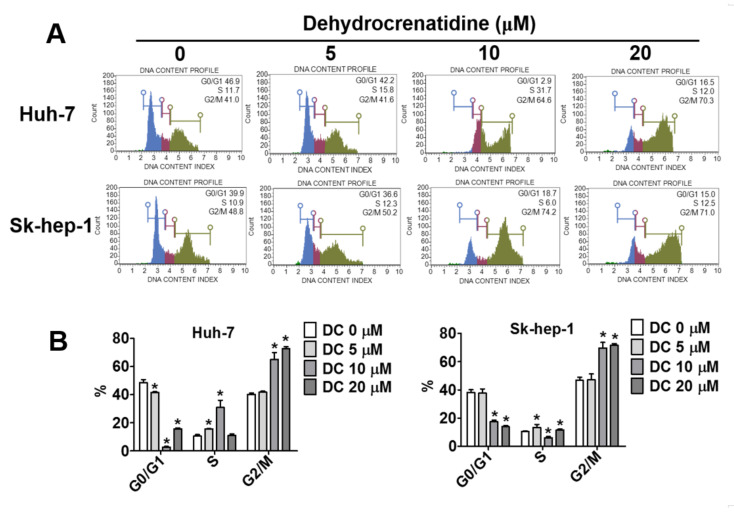
Dehydrocrenatidine induced cell cycle arrest in the G2/M phase in hepatoma cell lines. (**A**) The Huh-7 and Sk-hep-1 cells were assessed by flow cytometry after propidium iodide (PI) staining with indicated concentrations of 0, 5, 10, and 20 μM of Dehydrocrenatidine after 24 h. (**B**) Quantitative analysis of cell cycle data about G0/G1, S, G2/M of Huh-7 and Sk-hep-1 cells. Student’s *t* test was applied to determine statistical significance in three experiment. * represented *p* < 0.05, compared with the control group.

**Figure 3 pharmaceuticals-15-00402-f003:**
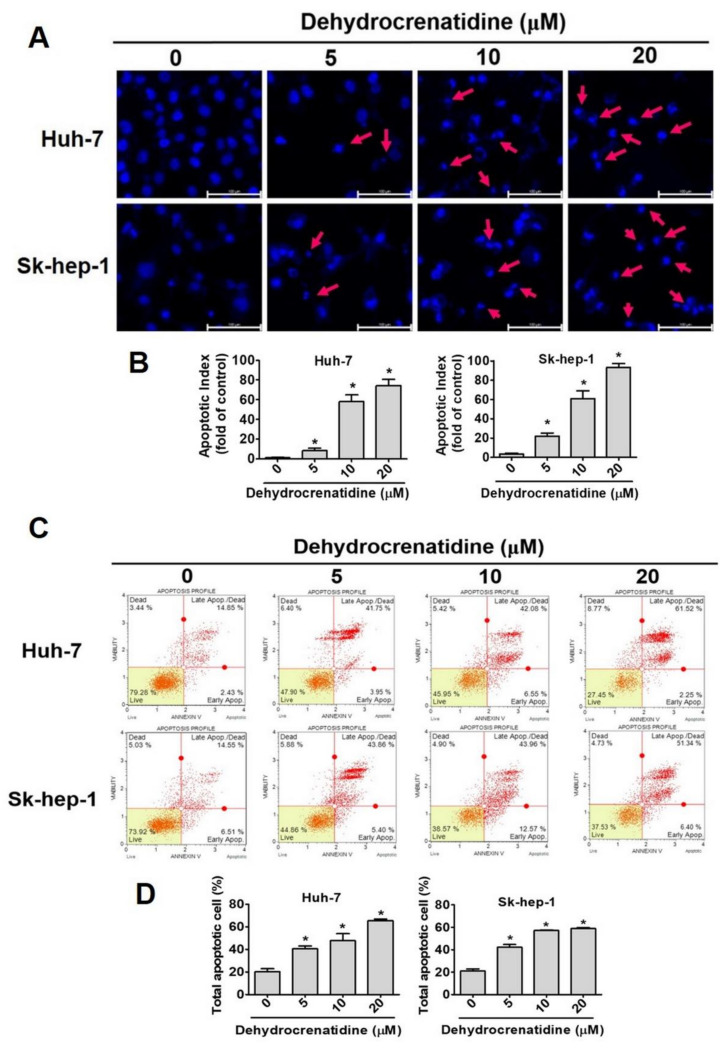
Dehydrocrenatidine induces apoptotic cell death in hepatoma cell lines. (**A**) The Huh-7 and Sk-hep-1 cells were stained with DAPI and observed under a fluorescence microscope. The red arrows show the nuclei condensation area. (**B**) The graphs show quantitative analysis of chromatin condensation of two cells. Bar scale = 100 µm. (**C**,**D**) After treated with indicated concentrations (0, 5, 10, and 20 μM) of dehydrocrenatidine, Huh-7 and Sk-hep-1 cells were stained with annexin-V and PI staining buffer and analyzed through flow cytometry. Student’s *t* test was applied to determine statistical significance in three experiment. * represented *p* < 0.05, compared with the control group.

**Figure 4 pharmaceuticals-15-00402-f004:**
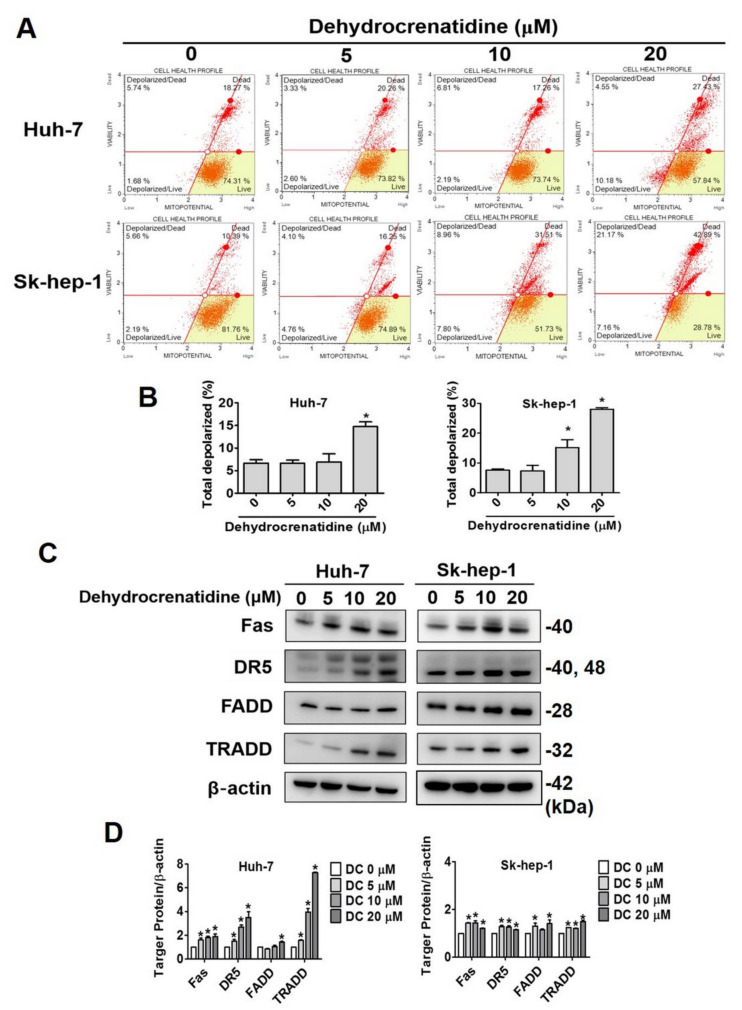
Dehydrocrenatidine activates apoptosis via the mitochondrial and death receptor pathways in hepatoma cell lines. (**A**,**B**) After treated with indicated concentrations of 0, 5, 10, and 20 μM of dehydrocrenatidine for 24 h, flow cytometry was used to analyze the effect on mitochondrial membrane potential and quantify the results of Huh-7 and Sk-hep-1 cells. (**C**) The death receptor related proteins were perfomed with Western blot to detect the protein expression of Fas, DR5, TRADD, FADD of Huh-7 and Sk-hep-1 cells. (**D**)The quantitative data of protein expression of two hepatoma cell lines was performed after being adjusted with β-actin. Student’s *t* test was applied to determine statistical significance in three experiment. * represented *p* < 0.05, compared with the control group.

**Figure 5 pharmaceuticals-15-00402-f005:**
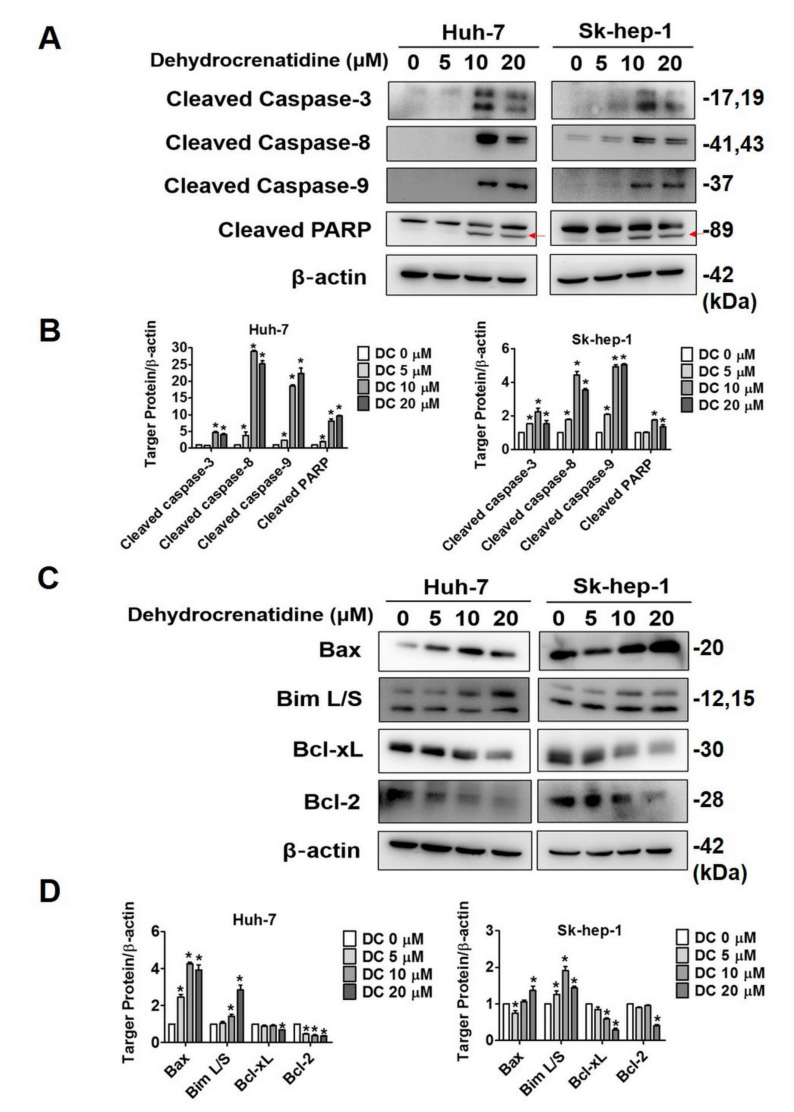
Dehydrocrenatidine increased apoptotic-related proteins and regulated pro-apoptotic proteins in hepatoma cell lines. (**A**,**B**) After treated with indicated concentrations of 0, 5, 10, and 20 μM of dehydrocrenatidine for 24 h, Western blot analysis detected an increasing on cleaved caspase-8, -9, -3, and cleaved PARP levels (red arrows). The protein expression of Huh-7 and Sk-hep-1 cell lines were quantified. (**C**,**D**) The protein expression of Bax, Bim L/S, Bcl-xL, Bcl-2 were detected after treatment of dehydrocrenatidine in Huh-7 and Sk-heap-1 cells, and quantified the protein expression of both cells. Student’s *t* test was applied to determine statistical significance in three experiment. * represented *p* < 0.05, compared with the control group.

**Figure 6 pharmaceuticals-15-00402-f006:**
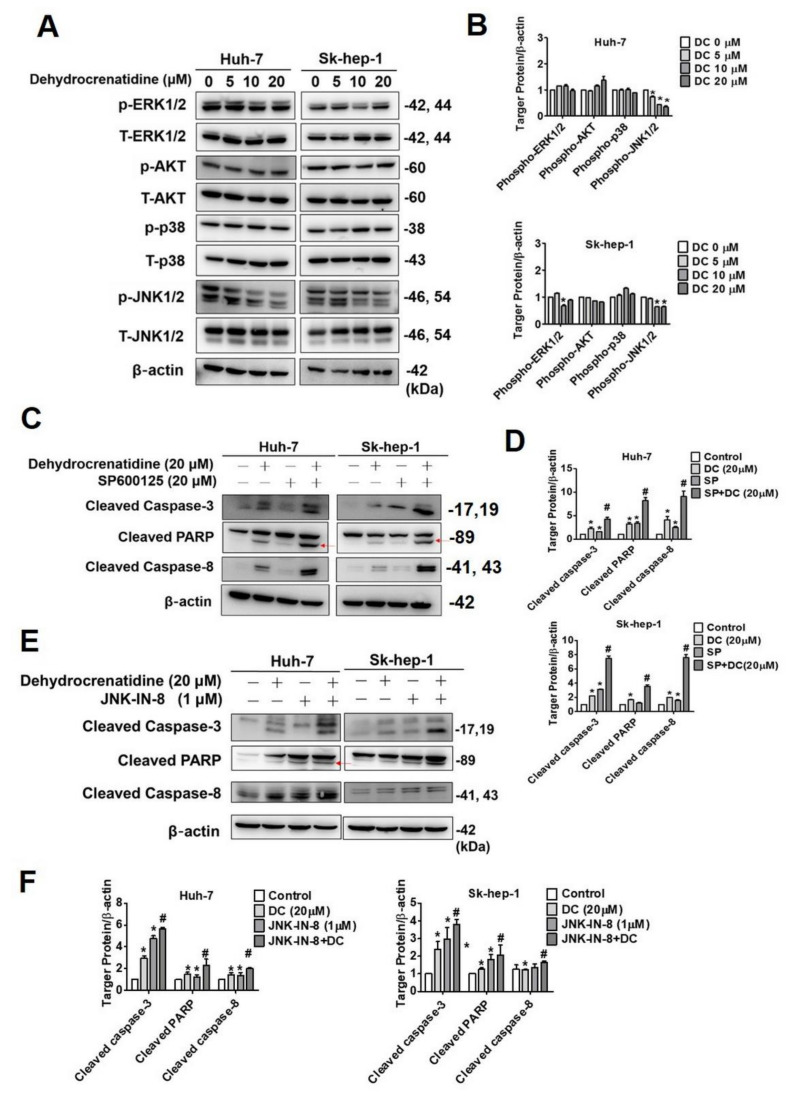
Dehydrocrenatidine inhibited the expression of phosphorylated JNK and regulated apoptosis process through MAPK pathways in hepatoma cell lines. (**A**) After treated with indicated concentrations of 0, 5, 10, and 20 μM of dehydrocrenatidine for 24 h, cell lysates were detected by Western blot with antibodies against of p-ERK1/2, p-AKT, p-p38, p-JNK1/2. (**B**) The graphs show quantitative analysis of protein expression of Huh-7 and Sk-hep-1 cell lines. (**C**) Huh-7 and Sk-hep-1 cells were pretreated with SP600125 for 1 h, and then incubated in the presence or absence of 20 μM dehydrocrenatidine for total 24 h. Cell lysates were detected by Western blot with antibodies against of cleaved caspase-8, -3, and PARP levels. (**D**) Quantification of protein expression levels in two hepatoma cell lines. (**E**) Specific JNK inhibitor, JNK-in-8, was involved in dehydrocrenatidine treatment (20 μM). The expression of cleaved caspase-8, -3, and cleaved PARP (red arrows) were detected by Western blot. (**F**) The graphs show quantitative analysis of protein expression of Huh-7 and Sk-hep-1 cell lines. All quantitative data of protein expression after being adjusted with β-actin. Student’s *t* test was applied to determine statistical significance in three experiment. * represented *p* < 0.05, compared with the control group. ^#^ represented *p* < 0.05, compared with the treated dehydrocrenatidine alone group.

## Data Availability

Data is contained within the article.

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
