# Peer review of "Dehydrocrenatidine Induces Liver Cancer Cell Apoptosis by Suppressing JNK-Mediated Signaling"

_pharmaceuticals, 2022, doi:10.3390/ph15040402_

Round 1

Reviewer 1 Report

The manuscript by Bharath Kumar et al. investigates the anticancer potential of the plant-based compound dehydrocrenatidine (DC). The authors find that DC inhibit the viability of two hepatic cancer cell lines by causing a G2/M arrest and inducing apoptosis. Then, they investigate the apoptotic pathways involved and find that DC induces apoptosis by inhibiting JNK phosphorylation.

Overall, the manuscript is well written, the results are mostly convincing and the conclusions are supported by the data presented. However, the quantification of Western Blots figures is questionable, and the involvement of JNK in DC-induced apoptosis must be further confirmed.

Major comment:

  • The statistical significance shown for western blots quantification from three different experiments is not convincing. Also, only actin blot is presented per figure even when several blots have been performed (as indicated by the fact that multiple proteins of same molecular weight are shown). The authors should provide in supplementary data the full western blots figures (including all actin blots) from the three experiments mentioned.
  • In Figure 6C, the conclusion of the authors that JNK inhibition by DC leads to apoptosis is only supported by the use of the compound sp600125, known to inhibit multiple kinases with similar or greater potency than JNK [PMID: 17850214]. Hence, these data should be confirmed by another JNK inhibitor and/or JNK silencing. Additionally, the effects on cell death should be confirmed, for instance by cell viability assay.

Minor comments:

  • The molecular weights of the detected bands should be indicated on the Western blots
  • The legend for Figure 1D is confusing and should be clarified.
  • The standard deviations/error bars are missing for the colony formation assay in Figure 1D.
  • In Figure 3a, it is impossible to see the chromatin condensation indicated by the authors due to the low magnification. Images at higher magnification should be shown.
  • In Figure 4c, the level of beta-actin for the Sk-hep-1 cells follows the level of the proteins analyzed which makes the conclusion unconvincing despite the quantification. Another representative image should be shown.

Reviewer 2 Report

Comment on the manuscript: "Dehydrocrenatidine induces liver cancer cell apoptosis by suppressing JNK-mediated signaling" by V. Bharath Kumar et al.

This study analyses the anticancer potential of the natural alkaloid dehydrocrenatidine in liver cancer cell. It reports several valuable results although it need a critical revision.

Abstract

The conclusion of the study: “dehydrocrenatidine reduced cancer cell viability by arresting cell cycle at G0/G1 phase” is not correct and is due to a cell cycle profile misinterpretation.

Figure legend 1:

It needs proofreading correction. Correct millimolar with micromolar.

Results

2.2. Effect of dehydrocrenatidine on cell cycle progression in liver cancer cells

To understand the anti-proliferative mode of action of dehydrocrenatidine, flow cytometric analysis was conducted to investigate the effects of different concentrations (5, 83 10 and 20 μM) of dehydrocrenatidine on the cell cycle progression of liver cancer cells. The findings revealed that the dehydrocrenatidine treatment of liver cancer cells caused significant cell cycle arrest at the G0/G1 phase. This statement is uncorrect, indeed a G0/G1 cell cycle arrest causes an increase of the corresponding peak.

"In contrast, a significant induction in cell cycle rate was observed at the G2/M phase following dehydrocrenatidine treatment (Figure 2)": This statement is incorrect because an increase in the G2/M peak/phase implies a G2/M cell cycle arrest not an induction of the cell cycle rate which should be estimated differently.

Surprisingly, the title of figure 2 is correct and is not coherent with what is written in the manuscript.

Figure 3 D

y-axis: apoptosos is uncorrect, please fix the mistake.

The authors should comment why Sk-hep-1 cells at 20 micromolar have less apoptotic cells  (Figure 2 D) and 100% of apoptotic index (Figure 2B). Are the y-axis values expressed as fold of increase or percentage of cells compared to control?

Figure 3 legend needs a grammar proofreading.

Figure 4 C and sentence:

The statement “The Western Blot findings revealed that the dehydrocrenatidine treatment significantly increased the expressions of death receptors 129 (FAS and DR5) and other death domain-containing adaptor proteins (FADD and TRADD) in liver cancer cells in a dose-dependent manner (Figure 4; C and D)”doesn’t exactly apply to Sk-hep-1 cells where the data are quite inconsistent. Again I’ m surprised because these cells should be at least equally sensitive to the alkaloid. My question is: are they affected by the same mechanism of cell death?

Data shown in Figure 5 and 6 are convincing.

Round 2

Reviewer 1 Report

The authors have adequately addressed most of my comments. I have no further concerns.

Reviewer 2 Report

The revision of the manuscript has addressed all the points raised by this reviewer and therefore it is now acceptable for publication.